# Smoothed Gradients for Stochastic Variational Inference

**Stephan Mandt**
Department of Physics
Princeton University
smandt@princeton.edu

**David Blei**
Department of Computer Science
Department of Statistics
Columbia University
david.blei@columbia.edu

## Abstract

Stochastic variational inference (SVI) lets us scale up Bayesian computation to massive data. It uses stochastic optimization to fit a variational distribution, following easy-to-compute noisy natural gradients. As with most traditional stochastic optimization methods, SVI takes precautions to use unbiased stochastic gradients whose expectations are equal to the true gradients. In this paper, we explore the idea of following biased stochastic gradients in SVI. Our method replaces the natural gradient with a similarly constructed vector that uses a fixed-window moving average of some of its previous terms. We will demonstrate the many advantages of this technique. First, its computational cost is the same as for SVI and storage requirements only multiply by a constant factor. Second, it enjoys significant variance reduction over the unbiased estimates, smaller bias than averaged gradients, and leads to smaller mean-squared error against the full gradient. We test our method on latent Dirichlet allocation with three large corpora.

## 1 Introduction

Stochastic variational inference (SVI) lets us scale up Bayesian computation to massive data [1]. SVI has been applied to many types of models, including topic models [1], probabilistic factorization [2], statistical network analysis [3, 4], and Gaussian processes [5].

SVI uses stochastic optimization [6] to fit a variational distribution, following easy-to-compute noisy natural gradients that come from repeatedly subsampling from the large data set. As with most traditional stochastic optimization methods, SVI takes precautions to use unbiased, noisy gradients whose expectations are equal to the true gradients. This is necessary for the conditions of [6] to apply, and guarantees that SVI climbs to a local optimum of the variational objective. Innovations on SVI, such as subsampling from data non-uniformly [2] or using control variates [7, 8], have maintained the unbiasedness of the noisy gradient.

In this paper, we explore the idea of following a biased stochastic gradient in SVI. We are inspired by the recent work in stochastic optimization that uses biased gradients. For example, stochastic averaged gradients (SAG) iteratively updates only a subset of terms in the full gradient [9]; averaged gradients (AG) follows the average of the sequence of stochastic gradients [10]. These methods lead to faster convergence on many problems.

However, SAG and AG are not immediately applicable to SVI. First, SAG requires storing all of the terms of the gradient. In most applications of SVI there is a term for each data point, and avoiding such storage is one of the motivations for using the algorithm. Second, the SVI update has a form where we update the variational parameter with a convex combination of the previous parameter and a new noisy version of it. This property falls out of the special structure of the gradient of the variational objective, and has the significant advantage of keeping the parameter in its feasible

space. (E.g., the parameter may be constrained to be positive or even on the simplex.) Averaged gradients, as we show below, do not enjoy this property. Thus, we develop a new method to form biased gradients in SVI.

To understand our method, we must briefly explain the special structure of the SVI stochastic natural gradient. At any iteration of SVI, we have a current estimate of the variational parameter $\lambda_i$, i.e., the parameter governing an approximate posterior that we are trying to estimate. First, we sample a data point $w_i$. Then, we use the current estimate of variational parameters to compute expected sufficient statistics $\hat{S}_i$ about that data point. (The sufficient statistics $\hat{S}_i$ is a vector of the same dimension as $\lambda_i$.) Finally, we form the stochastic natural gradient of the variational objective $\mathcal{L}$ with this simple expression:

$$\nabla_\lambda \mathcal{L} = \eta + N\hat{S}_i - \lambda_i, \tag{1}$$

where $\eta$ is a prior from the model and $N$ is an appropriate scaling. This is an unbiased noisy gradient [11, 1], and we follow it with a step size $\rho_i$ that decreases across iterations [6]. Because of its algebraic structure, each step amounts to taking a weighted average,

$$\lambda_{i+1} = (1 - \rho_i)\lambda_i + \rho_i(\eta + N\hat{S}_i). \tag{2}$$

Note that this keeps $\lambda_i$ in its feasible set.

With these details in mind, we can now describe our method. Our method replaces the natural gradient in Eq. (1) with a similarly constructed vector that uses a fixed-window moving average of the previous sufficient statistics. That is, we replace the sufficient statistics with an appropriate scaled sum, $\sum_{j=0}^{L-1} \hat{S}_{i-j}$. Note this is different from averaging the gradients, which also involves the current iteration's estimate.

We will demonstrate the many advantages of this technique. First, its computational cost is the same as for SVI and storage requirements only multiply by a constant factor (the window length $L$). Second, it enjoys significant variance reduction over the unbiased estimates, smaller bias than averaged gradients, and leads to smaller mean-squared error against the full gradient. Finally, we tested our method on latent Dirichlet allocation with three large corpora. We found it leads to faster convergence and better local optima.

**Related work**   We first discuss the related work from the SVI literature. Both Ref. [8] and Ref. [7] introduce control variates to reduce the gradient's variance. The method leads to unbiased gradient estimates. On the other hand, every few hundred iterations, an entire pass through the data set is necessary, which makes the performance and expenses of the method depend on the size of the data set. Ref. [12] develops a method to pre-select documents according to their influence on the global update. For large data sets, however, it also suffers from high storage requirements. In the stochastic optimization literature, we have already discussed SAG [9] and AG [10]. Similarly, Ref. [13] introduces an exponentially fading momentum term. It too suffers from the issues of SAG and AG, mentioned above.

## 2   Smoothed stochastic gradients for SVI

**Latent Dirichlet Allocation and Variational Inference**   We start by reviewing stochastic variational inference for LDA [1, 14], a topic model that will be our running example. We are given a corpus of $D$ documents with words $w_{1:D,1:N}$. We want to infer $K$ hidden topics, defined as multinomial distributions over a vocabulary of size $V$. We define a multinomial parameter $\beta_{1:V,1:K}$, termed *the topics*. Each document $d$ is associated with a normalized vector of topic weights $\Theta_d$. Furthermore, each word $n$ in document $d$ has a topic assignment $z_{dn}$. This is a $K-$vector of binary entries, such that $z_{dn}^k = 1$ if word $n$ in document $d$ is assigned to topic $k$, and $z_{dn}^k = 0$ otherwise.

In the generative process, we first draw the topics from a Dirichlet, $\beta_k \sim \mathrm{Dirichlet}(\eta)$. For each document, we draw the topic weights, $\Theta_d \sim \mathrm{Dirichlet}(\alpha)$. Finally, for each word in the document, we draw an assignment $z_{dn} \sim \mathrm{Multinomial}(\Theta_d)$, and we draw the word from the assigned topic, $w_{dn} \sim \mathrm{Multinomial}(\beta_{z_{dn}})$. The model has the following joint probability distribution:

$$p(w, \beta, \Theta, z | \eta, \alpha) = \prod_{k=1}^{K} p(\beta_k | \eta) \prod_{d=1}^{D} p(\Theta_d | \alpha) \prod_{n=1}^{N} p(z_{dn} | \Theta_d) p(w_{dn} | \beta_{1:K}, z_{dn}) \tag{3}$$

Following [1], the topics $\beta$ are global parameters, shared among all documents. The assignments $z$ and topic proportions $\Theta$ are *local*, as they characterize a single document.

In variational inference [15], we approximate the posterior distribution,

$$p(\beta, \Theta, z|w) = \frac{p(\beta, \Theta, z, w)}{\sum_z \int d\beta d\Theta \; p(\beta, \Theta, z, w)}, \tag{4}$$

which is intractable to compute. The posterior is approximated by a factorized distribution,

$$q(\beta, \Theta, z) = q(\beta|\lambda) \left( \prod_{d=1}^{D} \prod_{n=1}^{N} q(z_{dn}|\phi_{dn}) \right) \left( \prod_{d=1}^{D} q(\Theta_d|\gamma_d) \right) \tag{5}$$

Here, $q(\beta|\lambda)$ and $q(\Theta_d|\gamma_d)$ are Dirichlet distributions, and $q(z_{dn}|\phi_{dn})$ are multinomials. The parameters $\lambda$, $\gamma$ and $\phi$ minimize the Kullback-Leibler (KL) divergence between the variational distribution and the posterior [16]. As shown in Refs. [1, 17], the objective to maximize is the *evidence lower bound* (ELBO),

$$\mathcal{L}(q) = \mathbb{E}_q[\log p(x, \beta, \Theta, z)] - \mathbb{E}_q[\log q(\beta, \Theta, z)]. \tag{6}$$

This is a lower bound on the marginal probability of the observations. It is a sensible objective function because, up to a constant, it is equal to the negative KL divergence between q and the posterior. Thus optimizing the ELBO with respect to q is equivalent to minimizing its KL divergence to the posterior.

In traditional variational methods, we iteratively update the local and global parameters. The local parameters are updated as described in [1, 17] . They are a function of the global parameters, so at iteration $i$ the local parameter is $\phi_{dn}(\lambda_i)$. We are interested in the global parameters. They are updated based on the (expected) *sufficient statistics* $S(\lambda_i)$,

$$S(\lambda_i) = \sum_{d \in \{1, ..., D\}} \sum_{n=1}^{N} \phi_{dn}(\lambda_i) \cdot \mathcal{W}_{dn}^T \tag{7}$$

$$\lambda_{i+1} = \eta + S(\lambda_i)$$

For fixed $d$ and $n$, the multinomial parameter $\phi_{dn}$ is K×1. The binary vector $\mathcal{W}_{dn}$ is V×1; it satisfies $\mathcal{W}_{dn}^v = 1$ if the word $n$ in document $d$ is $v$, and else contains only zeros. Hence, $S$ is K×V and therefore has the same dimension as $\lambda$. Alternating updates lead to convergence.

**Stochastic variational inference for LDA**  The computation of the sufficient statistics is inefficient because it involves a pass through the entire data set. In Stochastic Variational Inference for LDA [1, 14], it is approximated by stochastically sampling a "minibatch" $B_i \subset \{1, ..., D\}$ of $|B_i|$ documents, estimating $S$ on the basis of the minibatch, and scaling the result appropriately,

$$\hat{S}(\lambda_i, B_i) = \frac{D}{|B_i|} \sum_{d \in B_i} \sum_{n=1}^{N} \phi_{dn}(\lambda_i) \cdot \mathcal{W}_{dn}^T.$$

Because it depends on the minibatch, $\hat{S}_i = \hat{S}(\lambda_i, B_i)$ is now a random variable. We will denote variables that explicitly depend on the random minibatch $B_i$ at the current time $i$ by circumflexes, such as $\hat{g}$ and $\hat{S}$.

In SVI, we update $\lambda$ by admixing the random estimate of the sufficient statistics to the current value of $\lambda$. This involves a learning rate $\rho_i < 1$,

$$\lambda_{i+1} = (1 - \rho_i)\lambda_i + \rho_i(\eta + \hat{S}(\lambda_i, B_i)) \tag{8}$$

The case of $\rho = 1$ and $|B_i| = D$ corresponds to batch variational inference (when sampling without replacement) . For arbitrary $\rho$, this update is just stochastic gradient ascent, as a stochastic estimate of the natural gradient of the ELBO [1] is

$$\hat{g}(\lambda_i, B_i) = (\eta - \lambda_i) + \hat{S}(\lambda_i, B_i), \tag{9}$$

This interpretation opens the world of gradient smoothing techniques. Note that the above stochastic gradient is unbiased: its expectation value is the full gradient. However, it has a variance. The goal of this paper will be to reduce this variance at the expense of introducing a bias.

---

**Algorithm 1:** Smoothed stochastic gradients for Latent Dirichlet Allocation

---
**Input**: $D$ documents, minibatch size $B$, number of stored
sufficient statistics $L$, learning rate $\rho_t$, hyperparameters $\alpha, \eta$.
**Output**: Hidden variational parameters $\lambda, \phi, \gamma$.

1   Initialize $\lambda$ randomly and $\hat{g}_i^L = 0$.
2   Initialize empty queue $Q = \{\}$.
3   **for** $i = 0$ ***to*** $\infty$ **do**
4     Sample minibatch $\mathcal{B}_i \subset \{1, \ldots, D\}$ uniformly.
5     initialize $\gamma$
6     **repeat**
7       For $d \in \mathcal{B}_i$ and $n \in \{1, \ldots, N\}$ set
8       $\phi_{dn}^k \propto \exp(\mathbb{E}[\log \Theta_{dk}] + \mathbb{E}[\log \beta_{k,w_d}]), k \in \{1, \ldots, K\}$
9       $\gamma_d = \alpha + \sum_n \phi_{dn}$
10    **until** $\phi_{dn}$ and $\gamma_d$ converge.
11    For each topic $k$, calculate sufficient statistics for minibatch $\mathcal{B}_i$:
12    $\hat{S}_i = \frac{D}{|B_i|} \sum_{d \in \mathcal{B}_i} \sum_{n=1}^{N} \phi_{dn} \mathcal{W}_{dn}^T$
13    Add new sufficient statistic in front of queue Q:
14    $Q \leftarrow \{\hat{S}_i\} + Q$
15    Remove last element when length $L$ has been reached:
16    **if** $\text{length}(Q) > L$ **then**
17      $Q \leftarrow Q - \{\hat{S}_{i-L}\}$
18    **end**
19    Update $\lambda$, using stored sufficient statistics:
20    $\hat{S}_i^L \leftarrow \hat{S}_{i-1}^L + (\hat{S}_i - \hat{S}_{i-L})/L$
21    $\hat{g}_i^L \leftarrow (\eta - \lambda_i) + \hat{S}_i^L$
22    $\lambda_{t+1} = \lambda_t + \rho_t\, \hat{g}_t^L$.
23   **end**

---

**Smoothed stochastic gradients for SVI**    Noisy stochastic gradients can slow down the convergence of SVI or lead to convergence to bad local optima. Hence, we propose a smoothing scheme to reduce the variance of the noisy natural gradient. To this end, we average the sufficient statistics over the past $L$ iterations. Here is a sketch:

1. Uniformly sample a minibatch $B_i \subset \{1, \ldots, D\}$ of documents. Compute the local variational parameters $\phi$ from a given $\lambda_i$.

2. Compute the sufficient statistics $\hat{S}_i = \hat{S}(\phi(\lambda_i), B_i)$.

3. Store $\hat{S}_i$, along with the $L$ most recent sufficient statistics. Compute $\hat{S}_i^L = \frac{1}{L} \sum_{j=0}^{L-1} \hat{S}_{i-j}$ as their mean.

4. Compute the *smoothed stochastic gradient* according to

$$\hat{g}_i^L = (\eta - \lambda_i) + \hat{S}_i^L \tag{10}$$

5. Use the smoothed stochastic gradient to calculate $\lambda_{i+1}$. Repeat.

Details are in Algorithm 1. We now explore its properties. First, note that smoothing the sufficient statistics comes at almost no extra computational costs. In fact, the mean of the stored sufficient statistics does not explicitly have to be computed, but rather amounts to the update

$$\hat{S}_i^L \leftarrow \hat{S}_{i-1}^L + (\hat{S}_i - \hat{S}_{i-L})/L, \tag{11}$$

after which $\hat{S}_{i-L}$ is deleted. Storing the sufficient statistics can be expensive for large values of $L$: In the context of LDA involving the typical parameters $K = 10^2$ and $V = 10^4$, using $L = 10^2$ amounts to storing $10^8$ 64-bit floats which is in the Gigabyte range.

Note that when $L = 1$ we obtain stochastic variational inference (SVI) in its basic form. This includes deterministic variational inference for $L = 1, B = D$ in the case of sampling without replacement within the minibatch.

**Biased gradients**    Let us now investigate the algorithm theoretically. Note that the only noisy part in the stochastic gradient in Eq. (9) is the sufficient statistics. Averaging over $L$ stochastic sufficient statistics thus promises to reduce the noise in the gradient. We are interested in the effect of the additional parameter $L$.

When we average over the $L$ most recent sufficient statistics, we introduce a bias. As the variational parameters change during each iteration, the averaged sufficient statistics deviate in expectation from its current value. This induces biased gradients. In a nutshell, large values of $L$ will reduce the variance but increase the bias.

To better understand this tradeoff, we need to introduce some notation. We defined the stochastic gradient $\hat{g}_i = \hat{g}(\lambda_i, B_i)$ in Eq. (9) and refer to $g_i = \mathbb{E}_{B_i}[\hat{g}(\lambda_i, B_i)]$ as the *full* gradient (FG). We also defined the smoothed stochastic gradient $\hat{g}_i^L$ in Eq. (10). Now, we need to introduce an auxiliary variable, $g_i^L := (\eta - \lambda_i) + \frac{1}{L}\sum_{j=0}^{L-1} S_{i-j}$. This is the time-averaged *full* gradient. It involves the full sufficient statistics $S_i = S(\lambda_i)$ evaluated along the sequence $\lambda_1, \lambda_{2,\dots}$ generated by our algorithm.

We can expand the smoothed stochastic gradient into three terms:

$$\hat{g}_i^L = \underbrace{g_i}_{\text{FG}} + \underbrace{(g_i^L - g_i)}_{\text{bias}} + \underbrace{(\hat{g}_i^L - g_i^L)}_{\text{noise}} \tag{12}$$

This involves the full gradient (FG), a bias term and a stochastic noise term. We want to minimize the statistical error between the full gradient and the smoothed gradient by an optimal choice of $L$. We will show this the optimal choice is determined by a tradeoff between variance and bias.

For the following analysis, we need to compute expectation values with respect to realizations of our algorithm, which is a stochastic process that generates a sequence of $\lambda_i$'s. Those expectation values are denoted by $\mathbb{E}[\cdot]$. Notably, not only the minibatches $B_i$ are random variables under this expectation, but also the entire sequences $\lambda_1, \lambda_2, \dots$ . Therefore, one needs to keep in mind that even the full gradients $g_i = g(\lambda_i)$ are random variables and can be studied under this expectation.

We find that the mean squared error of the smoothed stochastic gradient dominantly decomposes into a mean squared bias and a noise term:

$$\mathbb{E}[(\hat{g}_i^L - g_i)^2] \approx \underbrace{\mathbb{E}[(\hat{g}_i^L - g_i^L)^2]}_{\text{variance}} + \underbrace{\mathbb{E}[(g_i^L - g_i)^2]}_{\text{mean squared bias}} \tag{13}$$

To see this, consider the mean squared error of the smoothed stochastic gradient with respect to the full gradient, $\mathbb{E}[(\hat{g}_i^L - g_i)^2]$, adding and subtracting $g_i^L$:

$$\mathbb{E}\left[(\hat{g}_i^L - g_i^L + g_i^L - g_i)^2\right] = \mathbb{E}\left[(\hat{g}_i^L - g_i^L)^2\right] + 2\mathbb{E}\left[(\hat{g}_i^L - g_i^L)(g_i^L - g_i)\right] + \mathbb{E}\left[(g_i^L - g_i)^2\right].$$

We encounter a cross-term, which we argue to be negligible. In defining $\Delta\hat{S}_i = (\hat{S}_i - S_i)$ we find that $(\hat{g}_i^L - g_i^L) = \frac{1}{L}\sum_{j=0}^{L-1}\Delta S_{i-j}$. Therefore,

$$\mathbb{E}\left[(\hat{g}_i^L - g_i^L)(g_i^L - g_i)\right] = \frac{1}{L}\sum_{j=0}^{L-1}\mathbb{E}\left[\Delta\hat{S}_{i-j}(g_i^L - g_i)\right].$$

The fluctuations of the sufficient statistics $\Delta\hat{S}_i$ is a random variable with mean zero, and the randomness of $(g_i^L - g_i)$ enters only via $\lambda_i$. One can assume a very small statistical correlation between those two terms, $\mathbb{E}\left[\Delta\hat{S}_{i-j}(g_i^L - g_i)\right] \approx \mathbb{E}\left[\Delta\hat{S}_{i-j}\right]\mathbb{E}\left[(g_i^L - g_i)\right] = 0$. Therefore, the cross-term can be expected to be negligible. We confirmed this fact empirically in our numerical experiments: the top row of Fig. 1 shows that the sum of squared bias and variance is barely distinguishable from the squared error.

By construction, all bias comes from the sufficient statistics:

$$\mathbb{E}[(g_i^L - g_i)^2] = \mathbb{E}\left[\left(\frac{1}{L}\sum_{j=0}^{L-1}(S_{i-j} - S_i)\right)^2\right]. \tag{14}$$

At this point, little can be said in general about the bias term, apart from the fact that it should shrink with the learning rate. We will explore it empirically in the next section. We now consider the variance term:

$$\mathbb{E}[(\hat{g}_i^L - g_i^L)^2] = \mathbb{E}\left[\left(\frac{1}{L}\sum_{j=0}^{L-1}\Delta\hat{S}_{i-j}\right)^2\right] = \frac{1}{L^2}\sum_{j=0}^{L-1}\mathbb{E}\left[(\Delta\hat{S}_{i-j})^2\right] = \frac{1}{L^2}\sum_{j=0}^{L-1}\mathbb{E}[(\hat{g}_{i-j} - g_{i-j})^2].$$

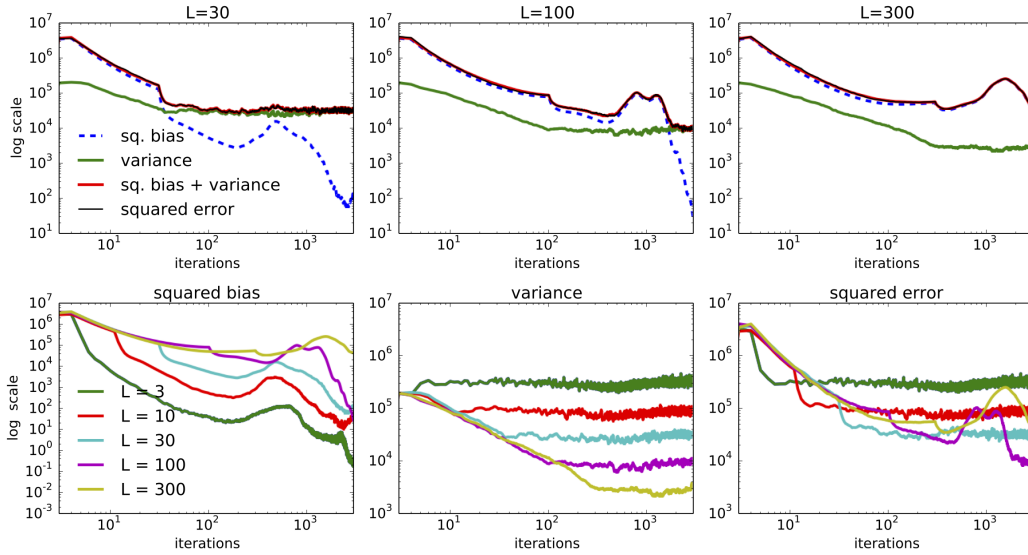

Figure 1: Empirical test of the variance-bias tradeoff on 2,000 abstracts from the Arxiv repository ($\rho = 0.01$, $B = 300$). **Top row.** For fixed $L = 30$ (left), $L = 100$ (middle), and $L = 300$ (right), we compare the squared bias, variance, variance+bias and the squared error as a function of iterations. Depending on $L$, the variance or the bias give the dominant contribution to the error. **Bottom row.** Squared bias (left), variance (middle) and squared error (right) for different values of $L$. Intermediate values of $L$ lead to the smallest squared error and hence to the best tradeoff between small variance and small bias.

This can be reformulated as $\text{var}(\hat{g}_i^L) = \frac{1}{L^2} \sum_{j=0}^{L-1} \text{var}(\hat{g}_{i-j})$. Assuming that the variance changes little during those $L$ successive updates, we can approximate $\text{var}(\hat{g}_{i-j}) \approx \text{var}(\hat{g}_i)$, which yields

$$\text{var}(\hat{g}_i^L) \quad \approx \quad \frac{1}{L}\text{var}(\hat{g}_i). \tag{15}$$

The smoothed gradient has therefore a variance that is approximately $L$ times smaller than the variance of the original stochastic gradient.

**Bias-variance tradeoff**  To understand and illustrate the effect of $L$ in our optimization problem, we used a small data set of 2000 abstracts from the Arxiv repository. This allowed us to compute the full sufficient statistics and the full gradient for reference. More details on the data set and the corresponding parameters will be given below.

We computed squared bias (SB), variance (VAR) and squared error (SE) according to Eq. (13) for a single stochastic optimization run. More explicitly,

$$\text{SB}_i = \sum_{k=1}^{K} \sum_{v=1}^{V} \left(g_i^L - g_i\right)_{kv}^2, \, \text{VAR}_i = \sum_{k=1}^{K} \sum_{v=1}^{V} \left(\hat{g}_i^L - g_i^L\right)_{kv}^2, \, \text{SE}_i = \sum_{k=1}^{K} \sum_{v=1}^{V} \left(\hat{g}_i^L - g_i\right)_{kv}^2. \tag{16}$$

In Fig. 1, we plot those quantities as a function of iteration steps (time). As argued before, we arrive at a drastic variance reduction (bottom, middle) when choosing large values of $L$.

In contrast, the squared bias (bottom, left) typically increases with $L$. The bias shows a complex time-evolution as it maintains memory of $L$ previous steps. For example, the kinks in the bias curves (bottom, left) occur at times $3, 10, 30, 100$ and $300$, i.e. they correspond to the values of $L$. Those are the times from which on the smoothed gradient looses memory of its initial state, typically carrying a large bias. The variances become approximately stationary at iteration $L$ (bottom, middle). Those are the times where the initialization process ends and the queue $Q$ in Algorithm 1 has reached its maximal length $L$. The squared error (bottom, right) is to a good approximation just the sum of squared bias and variance. This is also shown in the top panel of Fig. 1.

Due to the long-time memory of the smoothed gradients, one can associate some "inertia" or "momentum" to each value of $L$. The larger $L$, the smaller the variance and the larger the inertia. In a non-convex optimization setup with many local optima as in our case, too much inertia can be harmful. This effect can be seen for the $L = 100$ and $L = 300$ runs in Fig. 1 (bottom), where the mean squared bias and error curves bend upwards at long times. Think of a marble rolling in a wavy landscape: with too much momentum it runs the danger of passing through a good optimum and eventually getting trapped in a bad local optimum. This picture suggests that the optimal value of $L$ depends on the "ruggedness" of the potential landscape of the optimization problem at hand. Our empirical study suggest that choosing $L$ between 10 and 100 produces the smallest mean squared error.

**Aside: connection to gradient averaging**   Our algorithm was inspired by various gradient averaging schemes. However, we cannot easily used averaged gradients in SVI. To see the drawbacks of gradient averaging, let us consider $L$ stochastic gradients $\hat{g}_i, \hat{g}_{i-1}, \hat{g}_{i-2}, ..., \hat{g}_{i-L+1}$ and replace

$$\hat{g}_i \quad \longrightarrow \quad \tfrac{1}{L} \sum_{j=0}^{L-1} \hat{g}_{i-j}. \tag{17}$$

One arrives at the following parameter update for $\lambda_i$:

$$\lambda_{i+1} = (1 - \rho_i)\lambda_i + \rho_i \left( \eta + \frac{1}{L} \sum_{j=0}^{L-1} \hat{S}_{i-j} - \frac{1}{L} \sum_{j=0}^{L-1} (\lambda_{i-j} - \lambda_i) \right). \tag{18}$$

This update can lead to the violation of optimization constraints, namely to a negative variational parameter $\lambda$. Note that for $L = 1$ (the case of SVI), the third term is zero, guaranteeing positivity of the update. This is no longer guaranteed for $L > 1$, and the gradient updates will eventually become negative. We found this in practice. Furthermore, we find that there is an extra contribution to the bias compared to Eq. (14),

$$\mathbb{E}[(g_i^L - g_i)^2] \quad = \quad \mathbb{E}\left[ \left( \tfrac{1}{L} \sum_{j=0}^{L-1} (\lambda_i - \lambda_{i-j}) + \tfrac{1}{L} \sum_{j=0}^{L-1} (S_{i-j} - S_i) \right)^2 \right]. \tag{19}$$

Hence, the averaged gradient carries an additional bias in $\lambda$ - it is the same term that may violate optimization constraints. In contrast, the variance of the averaged gradient is the same as the variance of the smoothed gradient. Compared to gradient averaging, the smoothed gradient has a smaller bias while profiting from the same variance reduction.

## 3   Empirical study

We tested SVI for LDA, using the smoothed stochastic gradients, on three large corpora:

- 882K scientific abstracts from the Arxiv repository, using a vocabulary of 14K words.
- 1.7M articles from the New York Times, using a vocabulary of 8K words.
- 3.6M articles from Wikipedia, using a vocabulary of 7.7K words.

We set the minibatch size to $B = 300$ and furthermore set the number of topics to $K = 100$, and the hyper-parameters $\alpha = \eta = 0.5$. We fixed the learning rate to $\rho = 10^{-3}$. We also compared our results to a decreasing learning rate and found the same behavior.

For a quantitative test of model fitness, we evaluate the *predictive probability* over the vocabulary [1]. To this end, we separate a test set from the training set. This test set is furthermore split into two parts: half of it is used to obtain the local variational parameters (i.e. the topic proportions by fitting LDA with the fixed global parameters $\lambda$. The second part is used to compute the likelihoods of the contained words:

$$p(w_{\text{new}}|w_{\text{old}}, D) \approx \int \left( \sum_{k=1}^{K} \Theta_k \beta_{k,w_{\text{new}}} \right) q(\Theta)q(\beta)d\Theta d\beta = \mathbb{E}_q[\theta_k]\mathbb{E}_q[\beta_{k,w_{\text{new}}}]. \tag{20}$$

We show the predictive probabilities as a function of effective passes through the data set in Fig. 2 for the New York Times, Arxiv, and Wikipedia corpus, respectively. Effective passes through the data set are defined as (minibatch size * iterations / size of corpus). Within each plot, we compare

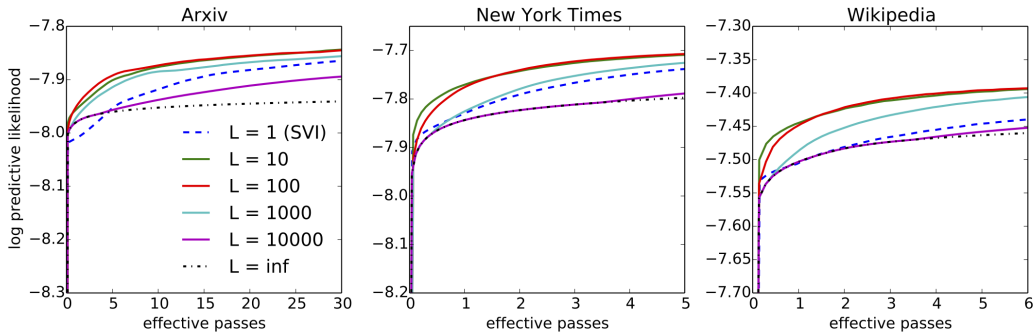

Figure 2: Per-word predictive probabilitiy as a function of the effective number of passes through the data (minibatch size * iterations / size of corpus). We compare results for the New York Times, Arxiv, and Wikipedia data sets. Each plot shows data for different values of $L$. We used a constant learning rate of $10^{-3}$, and set a time budget of 24 hours. Highest likelihoods are obtained for $L$ between 10 and 100, after which strong bias effects set in.

different numbers of stored sufficient statistics, $L \in \{1, 10, 100, 1000, 10000, \infty\}$. The last value of $L = \infty$ corresponds to a version of the algorithm where we average over *all* previous sufficient statistics, which is related to averaged gradients (AG), but which has a bias too large to compete with small and finite values of $L$. The maximal values of 30, 5 and 6 effective passes through the Arxiv, New York Times and Wikipedia data sets, respectively, approximately correspond to a run time of 24 hours, which we set as a hard cutoff in our study.

We obtain the highest held-out likelihoods for intermediate values of $L$. E.g., averaging only over 10 subsequent sufficient statistics results in much faster convergence and higher likelihoods at very little extra storage costs. As we discussed above, we attribute this fact to the best tradeoff between variance and bias.

## 4   Discussion and Conclusions

SVI scales up Bayesian inference, but suffers from noisy stochastic gradients. To reduce the mean squared error relative to the full gradient, we averaged the sufficient statistics of SVI successively over $L$ iteration steps. The resulting smoothed gradient is biased, however, and the performance of the method is governed by the competition between bias and variance. We argued theoretically and showed empirically that intermediate values of the number of stored sufficient statistics $L$ give the highest held-out likelihoods.

Proving convergence for our algorithm is still an open problem, which is non-trivial especially because the variational objective is non-convex. To guarantee convergence, however, we can simply phase out our algorithm and reduce the number of stored gradients to one as we get close to convergence. At this point, we recover SVI.

**Acknowledgements**   We thank Laurent Charlin, Alp Kucukelbir, Prem Gopolan, Rajesh Ranganath, Linpeng Tang, Neil Houlsby, Marius Kloft, and Matthew Hoffman for discussions. We acknowledge financial support by NSF CAREER NSF IIS-0745520, NSF BIGDATA NSF IIS-1247664, NSF NEURO NSF IIS-1009542, ONR N00014-11-1-0651, the Alfred P. Sloan foundation, DARPA FA8750-14-2-0009 and the NSF MRSEC program through the Princeton Center for Complex Materials Fellowship (DMR-0819860).

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
