[Reviews · NeurIPS 2014]

Submitted by Assigned_Reviewer_9

Summary
The paper introduces a simple strategy to reduce the variance of gradients in stochastic variational inference methods. Variance reduction is achieved by storing the last L data-point's contribution to the approximated/stochastic gradient and averaging these values. There exists a bias variance trade off : variance reduction comes at the cost of increased bias in the gradient estimates. The bias-variance tradeoff can be controlled by varying the sliding window size L. Also this strategy requires storing the last L data-point gradient contributions which can be significant.

Whilst it is not exact, I am convinced by the argument they present for their bias variance decomposition. It also appears to be supported emprically for the case of the LDA model they implemented. Maybe it would be clearer to explicitly plot the bias, the variance, bias + variance, and the mean squared error on the same axis (instead of the three figures presented in Fig (2)) as this may make the relation clearer.

Quality
The quality of the paper is high. The contribution is relatively simple however i believe it is an important one.

I would have appreciated more thorough experiments. These results should be over multiple runs? Or results from few different problems. These results are not convincing to me.

Clarity
The paper is extremely well written, there are hardly any typos, their presentation is thoughtful and considered.

I think you need to place footnote 1 in the main body of the paper. It feels out of place with the rest of the presentation.

Significance.
Whilst this is an incremental improvement to SVI methods I think because of the importance of this model class, and because scalability is so important, this is a simple yet significant contribution that would be of use to many in our community.

Minor points / typos:
line 671 - missing a the?
line 162 - admixing or just mixing
line 240 - equation reference error
line 260 - consider the mean...
line 264 - A priori
I think footnote 1 should be in the main body of the paper
Figure 2 - i think the text on the figures should be larger
eq (16) missing normalisation terms for these averages?
line 334 - we cannot easily used...
line 391 - probability spelling
Summary: Propose a simple averaging strategy to reduce variance at cost of increased bias in gradient estimates used in stochastic variational inference. Approach seems to increase convergence speed but this is neither proved or evidenced convincingly in the experiments section.

Submitted by Assigned_Reviewer_36

This paper proposes the smoothed gradient, which averages L recent sufficient statistics, to reduce the variance of the stochastic variational inference.

The proposed method is a straight-forward application of smoothed stochastic gradient and contains little technical depth.

p.5: Bias-variance trade-off of the proposed method is discussed. While the dependency of the variance on L is roughly identified, the bias is not analyzed. The analysis of the trade-off is finally resorted to simulation.

p.7: Although the empirical study handles large data sets, the application to two data sets is not comprehensive.

Minor:
p.3, l.109: as as
p.3, Eq.(3): The left hand side should indicate (\beta, \Theta,z,w).
p.3, l.126: The approximated distribution is (4) (not (3)). Is \Theta marginalized out?
p.3, l.154: minbatch -> minibatch
p.5, l.239: Equation number is missing.
p.5, l.264: A priory -> A priori
Summary: Although the proposed method improves the accuracy of stochastic variational inference, theoretical argument is insufficient, and the experiment is not so comprehensive.

Submitted by Assigned_Reviewer_40

Overview

The authors proposed a new approach of smoothing gradients for stochastic variational inference by averaging the sufficient statistics across a fixed time window. With the proposed method and a proper chosen time window size, the variance of stochastic gradients can be reduced, and the convergence speed of the algorithm can be improved. However, the sufficient statistics averaging introduce a bias to the gradients, which might lead to a less optimal solution.

Quality

This work proposes a gradient smoothing scheme via averaging the sufficient statistics. The gradient smoothing reduces variance of gradients, while paying the price by introducing a bias into gradients. It is an interesting idea to average the sufficient statistics instead of averaging gradients which has been studied before. The experiments with LDA show a good result empirically. It has been shown that the gradient smoothing scheme can reduce variance of gradients and improve model performance with a proper time-window size. The choice of the time-window size becomes a tuning parameter for the tradeoff between bias and variance. In general, introducing a small bias to the model is not a big problem, since variational inference approaches introduce an intrinsic bias while optimizing the evidence lower bound.

I have several reservations regarding this work. Different from introducing a bias into the objective function of model, the biases in gradients might be accumulated throughout learning, which potentially lead to a solution very far from local optima. It would be interesting to study the properties of the bias term, and see the new objective function that the bias term leads to. As mentioned in the discussion section, the convergence of the proposed method has not been proved. Some demonstration of convergence, at least empirically, needs to be shown. According to the experimental results, the choice of the time window size is crucially to the performance of the method. It would be useful to discuss the principal of choosing the right window size.

Clarity

This paper is, in general, well written, and easily understandable. The equation reference on line 240 is missing.

In figure 3, the results of SVI are not converged. Please show the results until convergence. It will be also good to show the performance of original LDA as a baseline, to have an idea of the performance difference between SVI-based approaches and original LDA for general audience.

Originality

This work looks like an original work with an interesting idea and good performance.

Significance

This work presents a new gradient smoothing scheme for stochastic variational inference. It would potentially be very useful for the variational inference community.
Summary: This work presents an interesting approach of smoothing stochastic gradients by averaging the sufficient statistics within a fixed time window. It is a new approach for smoothing gradients and the experiments shows a clear improvement over SVI. On the other hand, some more theoretical work needs to be done for further understanding it.
Author Feedback
Author rebuttal: We thank the reviewers for their constructive comments. In this
response, we would like to address some of their remarks.

1. R1 felt that "the proposed method is a straight-forward application
of smoothed stochastic gradient and contains little technical depth."

While we agree that our paper is based on a simple strategy which is
easy to implement, we disagree in that it is straightforward. As we
argue in the paper, a direct implementation of gradient averaging
[Nesterov, 2009] will generally fail in the context of SVI: the
averaged gradients can violate the constraints of the variational
parameters (as they do for LDA). Constrained optimization could
overcome this problem, but it is more complicated, computationally
demanding, and induces a bias larger than necessary (as we showed in
the paragraph "Aside: connection to gradient averaging").

In contrast, our smoothed gradients use the specific form of the
natural gradient in stochastic variational inference. Averaging only
parts of it (the sufficient statistics) satisfies the optimization
constraints and also induces less bias than in gradient averaging. In
general, stochastic variational inference can be used in many applied
settings (genetics, neuroscience, network analysis, Bayesian
nonparametrics, others). In all of these settings, our method can
exploit the special structure of its gradient.

2. R1 and R2 would have liked more theoretical analysis of the
algorithm.

We agree that it would be nice to have a better analytic understanding
of the bias term. However, such analysis is particularly difficult
and worthy of a research project on its own. The reason more theory
is difficult is because proofs of convergence of stochastic
optimization algorithms typically require Markovian dynamics and also
a convex objective. Neither applies in this context. In particular,
the bias term keeps long-term memory, which excludes the use of
semi-martingale techniques [Bottou, Online learning and stochastic approximations, 1998].

Even with the theoretical understanding left as future work, we feel
that our contributions here are significant: the new idea, the
interpretation via the bias/variance trade-off in the noisy gradient,
the empirical analysis of the bias, and the empirical study of overall
performance on large-scale data. These contributions, and the good
performance, point to it being useful to further study our method.

3. R1, R2 and R3 would have liked to see more empirical results on
large data sets, or on different problems/models.

We felt that two data sets demonstrate the idea well, especially
because they come from different domains (scientific papers and daily
news). We have also analyzed the original Wikipedia corpus from
Hoffman et al., 2010 and found the same performance. (We will add
this plot to the final version.) Further, with these encouraging
results, in our extended work we plan study the performance of
smoothed variational gradients on different models.

4. R2 is concerned that “the biases in gradients might be accumulated
throughout learning”. He also suggests “to study the properties of the
bias term, and see the new objective function that the bias term leads
to”.

R2 is correct that the bias can accumulate over iterations of the
algorithm. However, this accumulation is limited by the finite time
window L (and hence L should not be chosen too large in practice). If
theoretical convergence is a concern, the window L can be shrunk over
time to L=1. Once shrunk, the Robbins-Monro criteria guarantee
convergence to a local optimum for a properly decreasing learning rate
(Hoffman et al, 2013). Note that we did not find this to be necessary
in practice.

The idea to contemplate a new objective function is intriguing. However,
because it depends on the learning rate, we believe the bias cannot be derived
from a different objective function alone but is rather a “dynamic
property” of the optimization algorithm.

5. R2 wishes to see more empirical evidence for convergence and to “discuss
the principal of choosing the right window size [L]”.

This is a fair criticism and we plan to present more evidence of
convergence. In the submission we set a finite time budget, which is
also an interesting setting to consider.

As for the window size L, it will ultimately remain a tuning parameter
but a conservative choice of L=10, as we have shown empirically,
induces little bias while significantly reducing the variance. We
have run an extensive sensitivity study, which we will report on in
the final paper.

6. R2 proposes to “show the performance of original LDA as a baseline”

We thank the referee for this comment. The original SVI paper showed
SVI performing better than batch VI, but we agree that it is worth plotting
the original algorithm here as a baseline.

7. R3 points out that “storing the last L data-point gradient
contributions...can be significant”.

We agree. We also showed empirically that little extra memory costs
and extra bias are generated for e.g. L=10, while the variance
reduction is significant.

8. R3 proposes to “explicitly plot the bias, the variance, bias + variance,
and the mean squared error on the same axis”.

We thank the referee for this comment. We will add a corresponding
plot.

9. R3 proposes to “place footnote 1 in the main body of the paper”.

We agree and will integrate it in the main body.